REVIEW-SYMPOSIUM

# Astrocytes in functional recovery following central nervous system injuries

Qasim M. Alhadidi[1,2], Ghaith A. Bahader[3], Oiva Arvola[4,5], Philip Kitchen[6], Zahoor A. Shah[3] 
and Mootaz M. Salman[7,8]

[1]*Department of Anesthesiology, Perioperative and Pain Medicine, School of Medicine, Stanford University, Stanford, CA, USA*

[2]*Department of Pharmacy, Al-Yarmok University College, Diyala, Iraq*

[3]*Department of Medicinal and Biological Chemistry, College of Pharmacy and Pharmaceutical Sciences, University of Toledo, Toledo, OH, USA*

[4]*Division of Anaesthesiology, Jorvi Hospital, Department of Anaesthesiology, Intensive Care and Pain Medicine, University of Helsinki and Helsinki University Hospital, Helsinki, Finland*

[5]*Stem Cells and Metabolism Research Program, Research Programs Unit, Faculty of Medicine, University of Helsinki, Helsinki, Finland*

[6]*College of Health and Life Sciences, Aston University, Birmingham, UK*

[7]*Department of Physiology, Anatomy and Genetics, University of Oxford, Oxford, UK*

[8]*Kavli Institute for NanoScience Discovery, University of Oxford, Oxford, UK*

Handling Editors: Peying Fong & Robert Fenton

The peer review history is available in the Supporting Information section of this article (https://doi.org/10.1113/JP284197#support-information-section).

---

Q. M. Alhadidi and G. A. Bahader contributed equally to this work and share first authorship.

This article forms part of the 'Aquaporins in Health and Disease' symposium held at Copenhagen in September 2022, and organised by Robert Fenton.

The Journal of Physiology

**Abstract**   Astrocytes are increasingly recognised as partaking in complex homeostatic mechanisms critical for regulating neuronal plasticity following central nervous system (CNS) insults. Ischaemic stroke and traumatic brain injury are associated with high rates of disability and mortality. Depending on the context and type of injury, reactive astrocytes respond with diverse morphological, proliferative and functional changes collectively known as astrogliosis, which results in both pathogenic and protective effects. There is a large body of research on the negative consequences of astrogliosis following brain injuries. There is also growing interest in how astrogliosis might in some contexts be protective and help to limit the spread of the injury. However, little is known about how astrocytes contribute to the chronic functional recovery phase following traumatic and ischaemic brain insults. In this review, we explore the protective functions of astrocytes in various aspects of secondary brain injury such as oedema, inflammation and blood–brain barrier dysfunction. We also discuss the current knowledge on astrocyte contribution to tissue regeneration, including angiogenesis, neurogenesis, synaptogenesis, dendrogenesis and axogenesis. Finally, we discuss diverse astrocyte-related factors that, if selectively targeted, could form the basis of astrocyte-targeted therapeutic strategies to better address currently untreatable CNS disorders.

(Received 29 May 2023; accepted after revision 7 August 2023; first published online 13 September 2023)

**Corresponding author** M. M. Salman: Department of Physiology, Anatomy and Genetics, Kavli Institute for NanoScience Discovery, University of Oxford, Oxford, UK.    Email: mootaz.salman@dpag.ox.ac.uk

**Abstract figure legend** The roles of astrocytes in CNS injuries.

## Introduction

Brain injuries as a result of stroke and traumatic brain injury (TBI) cause functional and behavioural deficits and are a major cause of death and long-term disability worldwide. Globally, approximately 69 million people suffer from TBI and another 15 million suffer from strokes annually (Dewan et al 2018; Strong et al., 2007). The global annual cost of TBI has been estimated at $400 billion (Maas et al., 2017). Despite differences in the origin of the insult (exogenous in the case of TBI, endogenous in the case of stroke), the pathophysiology and the consequences of stroke and TBI are similar (Deb et al., 2010; Ng & Lee, 2019). Disability is one of the major consequences of brain injuries with about two-thirds of stroke survivors experiencing reduced mobility (Virani et al., 2020). With the increase in the survival rates of stroke patients from 56% in 1985 to 66% in 2016, subsequent disability is an increasingly serious public health concern that is associated with negative physical,

psychological and economic outcomes for patients, their families and society as a whole (Aked et al 2021; Zorowitz et al., 2009).

Brain injury disrupts neuronal circuits and can result in cognitive, sensory or motor deficits, depending on the location and the severity of the lesion (Wieloch & Nikolich, 2006). Cells in the lesion and peri-lesion areas undergo cell death and damage followed by a cascade of events including blood–brain barrier (BBB) disruption, oedema, inflammation, oxidative stress and excitotoxicity. Altogether, these processes are referred to as secondary brain injury as they can lead to further cell death, demyelination and axonal degeneration and create an inhibitory milieu for regeneration and recovery (Fawcett, 2006). Functional recovery starts soon after brain injury and involves the recovery of the function of the lesioned area by reorganisation of the adjacent non-lesioned area (Kolb et al., 2011; Wittenberg, 2010). This depends on the plastic nature of the brain and can take place over a long period (months to years)

**Mootaz Salman** is a Group Leader in Cellular Neuroscience and Research Lecturer in the Department of Physiology, Anatomy and Genetics (DPAG) at the University of Oxford, and in the Kavli Institute for Nanoscience Discovery in Oxford. His research focuses on glial cells (particularly astrocytes), neurodegenerative diseases, blood–brain barrier, 3D humanised models of brain-on-a-chip, patient-derived stem cells and genetic engineering (CRISPR-Cas9).

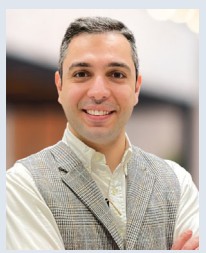

(Cassidy & Cramer, 2017). In general, the recovery process involves clearance of dead cell debris, resolution of inflammation and oedema, neurogenesis, gliogenesis, angiogenesis, cell repair (axonal sprouting, dendritic growth and synaptogenesis) and consolidation of new neuronal circuits (Wieloch & Nikolich, 2006). Clearance of cellular debris and resolution of inflammation and oedema by glial cells creates a facilitative environment for the recovery process. During ageing, the decline in the regenerative capacity of axons is not due to axonal intrinsic factors but instead to ageing glia and impairment in the clearance of myelin debris (Painter et al., 2014), suggesting a crucial role for glia in regeneration.

Microglia, astrocytes and oligodendrocytes are collectively known as glial cells. They provide physical, functional and metabolic support to neurons. Astrocytes regulate brain homeostasis and the extracellular environment (Ransom & Ransom, 2012). The finely branched astrocytic processes are classified, based on the morphological and functional properties, into the organelle-containing branches and organelle-free leaflets (Semyanov & Verkhratsky, 2021). Astrocytes contact all cellular components of the CNS and a single astrocyte can contact up to 2 million neuronal synapses in the human brain (Bushong et al., 2002; Halassa et al., 2007; Zhou et al., 2019). Furthermore, in most brain regions, every astrocyte directly contacts at least one blood vessel through its specialised and polarised endfeet (Hösli et al., 2022). Astrocytes regulate water and ion homeostasis using specific $Na^+$, $K^+$, $Cl^-$ and water channels (Min & van der Knaap, 2018), maintain neurotransmitter homeostasis by active reuptake of glutamate and GABA via glutamate and GABA transporters (Verkerke et al., 2021), maintain BBB integrity and regulate cerebral blood flow (Verkhratsky & Nedergaard, 2018). Astrocytes can also provide metabolic and trophic support to neurons via the astrocyte–neuron lactate shuttle and secretion of growth factors such as brain-derived neurotrophic factor (BDNF) (Bordone et al., 2019; Martin et al., 2013). Astrocytes produce antioxidant enzymes and molecules such as superoxide dismutase and glutathione and therefore contribute to CNS redox homeostasis (Chen, Qin et al., 2020; Salman et al., 2017). In cooperation with microglia, astrocytes regulate synapse formation and pruning and are actively involved in shaping neuronal circuits (Chung et al 2013; Kucukdereli et al., 2011).

Importantly, astrocytes in different brain regions can differ significantly in their transcriptomes, proteomes and phenotypic properties (Batiuk et al., 2020; Clarke et al., 2021). Although we refer to astrocytes generally throughout this review, we caution the reader that some results are likely to be brain region-specific and where possible we have identified specific brain regions in *in vivo* studies. Data from a recent single cell sequencing study demonstrating astrocyte regional heterogeneity is publicly available at https://holt-sc.glialab.org/sc/, which we encourage the reader to explore.

In response to brain injury, astrocytes undergo morphological and functional changes characterised by cellular hypertrophy, proliferation and increased expression of intermediate filament proteins such as glial fibrillary acidic protein (GFAP), a process commonly known as reactive astrogliosis (Pekny & Pekna, 2014). In recent years, growing evidence supports measurement of GFAP in cerebrospinal fluid or serum as a biomarker for diagnosis and prognosis of neurodegenerative diseases, stroke and TBI (Abdelhak et al., 2022). GFAP levels reflect the severity of brain injury in TBI patients and could help in discriminating patients with abnormal head CT scans from patients with normal findings (Czeiter et al., 2020; Diaz-Arrastia et al., 2014; Mahan et al., 2019).

The term 'astrogliopathy' refers to a group of disorders involving astrocytes that may have genetic or environmental origins, leading to aberrant cellular phenotypes and a variety of neurological conditions such as neurodegenerative diseases, psychiatric disorders or seizures (Verkhratsky et al., 2017). In general, astrogliopathies are divided into three types: (i) astrodegeneration with astroglial atrophy and loss of function; (ii) pathological remodelling of astrocytes; and (iii) reactive astrogliosis (Verkhratsky & Parpura 2016; Verkhratsky et al 2015).

Astrodegeneration, which leads to several neurological illnesses, including psychiatric and neurodegenerative diseases, manifests primarily as astroglial morphological atrophy, cell death and/or loss of function (Verkhratsky et al., 2017). Tissue homeostasis is severely compromised by astrocyte cell death, which reduces synaptic glutamate clearance, disrupts neurotransmission and can induce psychotic symptoms (Sanacora & Banasr 2013; Verkhratsky et al., 2014). In amyotrophic lateral sclerosis, astrodegeneration and impaired glutamate clearance cause excitotoxic degeneration of motor neurons leading to pathology (Rossi et al., 2008). Similarly, the decline of astrocyte synaptic coverage by astrodegeneration disrupts synaptic function and results in cognitive impairment, a hallmark of Alzheimer's disease (Verkhratsky et al., 2015). The primary cause of astrocyte pathological remodelling is the acquisition of aberrant function, which impairs the homeostatic functions of astrocytes. In Alexander disease, for example, GFAP mutations cause severe leukomalacia (Messing et al., 2012). The process of reactive astrogliosis is complex and depends on the nature and severity of the injury (Sofroniew, 2014b). For around a decade, researchers have debated whether astrogliosis is a reaction that is detrimental or beneficial to functional recovery (Sofroniew, 2014a). However, there is mounting evidence that astrogliosis can be either beneficial or detrimental depending on the nature of the function gained or lost following injury

(Escartin et al., 2021). Reactive astrogliosis is regarded as a defensive mechanism to restore tissue homeostasis. This is accomplished by a variety of mechanisms, including the provision of neuronal trophic support, restriction of the lesioned area, regeneration of damaged tissue and restoration of disrupted neural circuits (Anderson et al., 2016; Pekny & Pekna 2014; Pekny et al., 2016; Sofroniew, 2014b). In TBI and ischaemic stroke models, inhibition of astrogliosis or formation of the perilesional glial barrier (PLGB – sometime referred to as 'glial scar') exacerbated inflammation and brain injury and impaired functional recovery (Faulkner et al., 2004; Li et al., 2008b; Myer et al., 2006). Other studies, however, reported that astrogliosis aggravated inflammation and impeded axonal regeneration following brain injury (Silver & Miller 2004; Spence et al., 2011).

While the majority of reviews focus on the neuroprotective role of reactive astrocytes during the acute phase of injury, few discuss the role of astrocytes in the recovery process. Therefore, we describe here current understanding of the role of astrocytes in the recovery process following CNS injuries, highlighting prospective targets for recovery-promoting therapies.

## Mechanisms of functional recovery

Following brain injury, there is neuroplasticity in the form of cortical remapping in which the function of the damaged region is recovered by remapping to another part of the cortex (Wittenberg, 2010). The remapping process occurs concomitantly with the growth of new connections and involves the generation of new circuits beyond the lesioned area (Nagappan et al., 2020). Accordingly, neural recovery encompasses structural and functional changes which reorganise neuronal connections and result in the formation of new synapses through a variety of mechanisms including axonal sprouting, dendritic branching, neurogenesis, axon preservation, remyelination, re-establishment of BBB integrity, inhibition of extracellular inhibitory signals, alteration of excitability, and promotion of new cortical maps and neural networks (Regenhardt et al., 2020).

Axonal sprouting represents an essential part of the functional recovery process. New axons can connect locally within the peri-infarct area or over longer distances between different lobes of the damaged hemisphere (Brown et al., 2009; Thomas Carmichael et al., 2001). There is evidence of axonal sprouting following injury in humans, with the neuronal growth cone protein growth-associated protein-43 (GAP43) upregulated in the peri-infarct area (Sandelius et al., 2018). Stroke induces axonal sprouting from the contralateral cortex to parts of the cervical spinal cord and brainstem that have been denervated from the lesion site (Lindau et al., 2014; Wahl et al., 2014). In larger experimental stroke lesions such

as the photothrombotic stroke model with more damage in the ipsilateral cortex than in middle cerebral artery occlusion (MCAO) models, axonal sprouting from the contralateral cortical regions is associated with recovery (Bachmann et al., 2014; Wahl et al., 2014). Likewise, the promotion of axonal sprouting from the intact contralateral hemisphere into the denervated spinal cord enhances functional recovery in rats following TBI (Zhang et al., 2010). Dendritic branching is also likely involved in the recovery process. In a mouse photothrombotic stroke model, dendritic spine and synapse loss was reported in the peri-lesion area during the first week and was followed by branched growth and recovery at 2 weeks (Brown et al., 2010).

The molecular and cellular adaptations that support these structural changes include changes in gene expression and secretion of cytokines, chemokines and growth factors from glial and endothelial cells (Akide Ndunge et al., 2022; Davis et al 2019; Krakauer & Carmichael, 2017), and the cellular processes of neurogenesis, angiogenesis and gliogenesis (Imayoshi et al., 2008; Lee & Thuret, 2018). The generation of new neurons from neural stem cells (neuroblasts) has been demonstrated in rodent models of CNS ischaemia (Zhang et al., 2016). Newly generated neuroblasts, originating from the subgranular zone of the dentate gyrus or the subventricular zone (SVZ), migrate to the peri-ischaemic regions and differentiate into new neurons in response to brain injury. A large body of evidence demonstrates that augmenting or inhibiting neurogenesis in experimental stroke models is associated with improved (Leker et al., 2007; Ohab et al., 2006) or impaired (Imayoshi et al., 2008; Wang et al., 2012) recovery, respectively. Moreover, human biopsy and autopsy studies suggest that neuronal migration happens following a stroke (Jin et al., 2006). Interestingly, the originating cell type for these new neurons is not restricted to SVZ stem cells; they may also be derived from astrocytes and pericytes. It has been demonstrated that reactive astrocytes in the striatum can transdifferentiate into neuroblasts that develop into mature neurons within approximately 2 weeks in mouse stroke models (Duan et al., 2015; Shen et al., 2016), and similarly for brain pericytes (Nakata et al., 2017). These data indicate that new neurons from a variety of originating cell types are integrated into newly established circuits to support the brain repair process following ischaemic injury.

Angiogenesis contributes substantially to brain repair and functional recovery following brain injury. Newly formed blood vessels supply nutrients, provide a route for clearance of the breakdown products of damaged tissue and provide neurovascular niches for neuroblast migration following brain injury (Ergul et al., 2012). Angiogenesis is triggered relatively quickly following TBI or ischaemic injuries, with endothelial cell proliferation observable as early as 12 h after stroke in rodents (Beck

& Plate, 2009), primarily in the peri-infarct area, but also extending to other regions such as the contralateral cortex (Ergul et al., 2012; Liu et al., 2014). It is worth mentioning that most neurorestorative agents that improve functional outcomes after TBI and stroke are associated with increased neurogenesis and angiogenesis (Xiong et al., 2010).

Myelin loss and subsequent axonal dysfunction can occur following brain injury, which negatively impacts brain function. Oligodendrocytes, the main cell type responsible for producing myelin sheaths, cannot generate new myelin following ischaemic stroke and TBI, leaving axons vulnerable to further injury (Dent et al., 2015; Flygt et al., 2013; Sozmen et al., 2009). Remyelination during the brain repair process occurs following oligodendrogenesis, which involves the proliferation, migration, differentiation and maturation of new myelinating oligodendrocytes from oligodendrocyte precursor cells (OPCs) (Itoh et al., 2015; Maki et al., 2013). Newly generated oligodendrocytes have been reported in the peri-infarct area during the stroke recovery phase (56 days post-stroke) in mouse and rat MCAO models (Ueno et al 2012; Zhang et al., 2011). Moreover, the number of OPCs has been reported to be increased on day 2 and day 7 after injury in rats subjected to a fluid percussion model of TBI (Flygt et al., 2013).

Alteration of neuronal excitability has been identified in both peri-infarct areas and remote areas which are structurally and functionally connected to stroke or TBI infarct areas (Carrera & Tononi, 2014). Following a stroke, the peri-infarct area becomes hypoexcitable, mainly due to impaired uptake of the inhibitory neurotransmitter $\gamma$-aminobutyric acid (GABA), leading to increased tonic GABA signalling mediated by extrasynaptic GABA$_A$ receptors (Brickley & Mody, 2012). Moreover, the expression of monoamine oxidase B (MAO-B), a key enzyme for astrocytic GABA synthesis and GABA activity, was reported to be significantly increased in reactive astrocytes in rodent models of ischaemic stroke and Alzheimer's disease (Jo et al., 2014; Nam et al., 2020a). Downregulation of GABA transporters in reactive astrocytes is an important contributor to this potentiation of tonic GABA signalling in the peri-infarct area (Carmichael, 2012). Accordingly, the re-establishment of excitatory/inhibitory balance has been explored preclinically as a therapeutic strategy to promote functional recovery. This can be achieved pharmacologically by inhibition of tonic GABA$_A$ signalling or potentiation of excitatory glutamatergic signalling (Kumar & Kitago, 2019). Therapeutically, it would be important to time such an intervention appropriately, in order to avoid exacerbating glutamate-associated excitotoxicity in the acute phases of stroke and TBI. Inhibition of tonic GABA signalling has been shown to provide structural and functional benefits by augmenting long-term potentiation

and remapping sensorimotor functions in both hemispheres (Chen et al., 2011; Hagemann et al., 1998; Takatsuru et al., 2009). In mice, recovery of motor function was improved by delayed administration of an $\alpha$5-GABA$_A$ receptor antagonist (Clarkson et al., 2010). Similarly, administration of a positive allosteric modulator of AMPA receptors enhanced recovery of motor function by increasing BDNF expression in the peri-infarct region (Clarkson et al., 2011).

Extracellular matrix (ECM) proteins such as chondroitin sulfate proteoglycans (CSPGs) constitute the major component of the PLGB which is formed by peri-lesional reactive astrocytes following traumatic injury or stroke (Silver & Miller, 2004). CSPGs inhibit neurite outgrowth and axonal sprouting and are therefore considered a major cause of regenerative failure following CNS injury (Kwok et al., 2012). Chondroitinase ABC, which degrades CSPGs, promoted recovery of motor function following focal cortical ischaemia or spinal cord injury in rats when administered locally at the lesion site (Bradbury et al., 2002; Gherardini et al., 2015). Inhibition of the CSPG receptor PTP$\sigma$ either by gene deletion or a receptor inhibitory peptide in a mouse stroke model led to increased axonal sprouting and enhanced penetration of the PLGB by neural stem cells, which improved post-stroke motor and cognitive recovery (Luo et al., 2022). Similar results were reported following PTP$\sigma$ inhibition in a rat spinal cord injury model (Urban et al., 2020), and inhibition of the related CSPG receptor LAR also led to similar improvements in axonal sprouting and functional recovery (Cheng et al., 2021). Intriguingly, PTP$\sigma$ and LAR appear to inhibit axonal sprouting by different downstream mechanisms (Sami et al., 2020), suggesting the potential for additive or synergistic effects of dual inhibition (Ohtake et al., 2016). Semaphorins and ephrins are growth cone inhibitors that inhibit the extension of neurites and impede regeneration. In a mouse photothrombotic stroke model, reactive astrocytes in the peri-infarct cortex had significant upregulated ephrin-A5, which inhibited axonal sprouting and motor recovery. Blockage of ephrin-A5 signalling by a soluble receptor decoy (EphA5-Fc) increased axonal sprouting within the peri-infarct region, facilitated cortical remapping and improved recovery of motor function (Overman et al., 2012).

## Astrocytes in oedema and inflammation

Cerebral oedema is associated with the development of symptoms in various neurological disorders and is associated with increased mortality and morbidity (Nawabi et al., 2019). Stroke and TBI are characterised by multiple stages of cerebral oedema in the acute and chronic phases following the injury, with most of the initiating mechanisms related to astrocytes (Fig. 1).

Cytotoxic oedema is the first stage of cerebral oedema and occurs a few hours after the ischaemic event and as early as 1 h post-TBI in humans (Ito et al., 1996; Salman, Kitchen, Halsey, Wang & Tornroth-Horsefield, 2021). In cytotoxic oedema, the astrocytic swelling occurs due to ATP depletion and the influx of extracellular ions such as $Na^+$ and $Cl^-$ down their electrochemical gradients leading to elevated intracellular osmolarity that drives water flux into astrocytes without immediately affecting the brain tissue volume (Kitchen et al 2020; Michinaga & Koyama 2015b; Sylvain et al., 2021). The resultant increase in intracellular $[Na^+]$ and decrease in extracellular $[Na^+]$ disrupts the ionic equilibrium across the BBB, increasing the flux of water, $Na^+$ and other ions from blood and perivascular spaces and through several astrocyte and endothelial channel proteins and transporters, which consequently initiates ionic oedema (Mestre et al., 2020; Mori et al., 2002; Stiefel et al., 2005). Finally, during the later phase of ischaemic stroke (24−48 h after onset) (Dostovic et al., 2016), BBB disruption leads to the extravasation of water and intravascular solutes, including serum proteins, which cause vasogenic oedema, increase intracranial pressure and exacerbate brain damage (Michinaga & Koyama 2015b).

Astrocytes have a major role in the formation and resolution of cerebral oedema. $Na^+$ signalling and homeostasis control numerous astrocyte-mediated CNS functions and responses through controlling various transporters that are important for ion and neurotransmitter homeostasis (Rose & Verkhratsky, 2016). $Na^+–K^+–Cl^-$–co-transporter 1 (NKCC1) is expressed in astrocytes and is a key facilitator of cytotoxic and ionic oedema (Su et al., 2000). Expression of NKCC1 was increased in a rat TBI model (Dostovic et al., 2016), in fluid percussion-injured cultured rat astrocytes (Mestre et al., 2020) and in cultured human astrocytes exposed to high glucose (Su et al., 2000). This would be expected to increase sodium, chloride and potassium influx (Jayakumar et al., 2011), although as NKCC1 is a sodium-driven secondary active transporter, as the $Na^+/K^+$-ATPase fails and $[Na^+]_i$ increases, the driving force for NKCC1 would be expected to decrease. Treatment with bumetanide, an inhibitor of NKCC1 or knockout of NKCC1 in mice both mitigate the cytotoxic oedema induced by TBI and cerebral ischaemia in the acute and chronic stages (Chen et al., 2005; Lu et al., 2008; Wang, Huang, He, Ruan & Huang, 2014a; Xu et al., 2016, 2017).

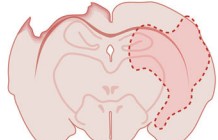

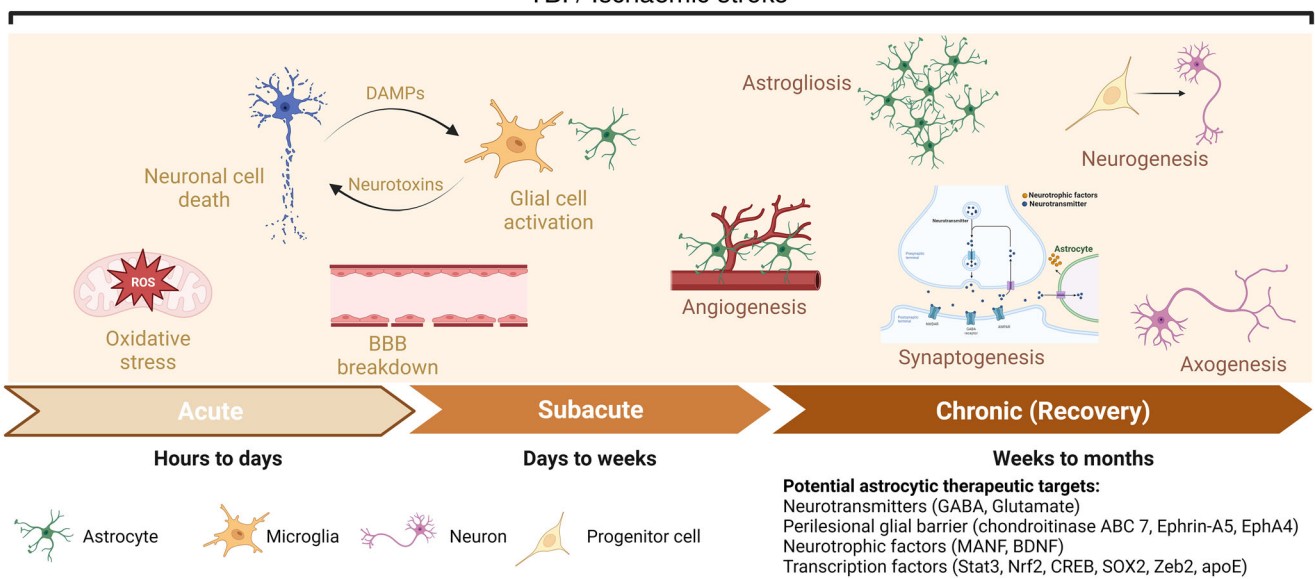

**Figure 1. Schematic timeline of processes after TBI or stroke**
The acute and subacute phases are characterised by distinct pathological mechanisms with oxidative stress, neuro-inflammation represented by glial cell activation and cytokine release, and blood–brain barrier (BBB) disruption as key elements that can be targeted in this early stage. The chronic phase involves recovery processes including angiogenesis, synaptogenesis, axogenesis, astrogliosis and neurogenesis. Promoting these processes through potential astrocyte-related targets during this phase may be an effective therapeutic strategy.

The sulfonylurea receptor 1–transient receptor potential melatonin 4 (SUR1–TRPM4) complex is another important astrocytic channel expressed following CNS injury and facilitates cytotoxic oedema formation (Salman, Kitchen, Yool & Bill, 2021; Stokum et al., 2018). The activation of the HIF-1 transcription factor (TF) following hypoxia and CNS injury triggers the transcription of the *ABCC1* gene encoding the SUR1 protein, which binds TRPM4 and makes it more sensitive to ATP (Gu et al., 2022; Salman et al 2017). Following ischaemia or TBI, ATP depletion results in the opening of the nucleotide-gated SUR1–TRPM4 channel, sodium influx, membrane depolarisation and induction of cyto-toxic oedema (Simard et al., 2006; Simard, Kilbourne et al., 2009). Inhibition of the SUR1–TRPM4 channel has been suggested as a therapeutic target for cytotoxic oedema following cerebral ischaemic and traumatic brain injuries. Pharmacological inhibition of SUR1–TRPM4 with the SUR1 inhibitor glibenclamide reduced brain oedema and associated brain damage in several rat models of ischaemic stroke and TBI (Simard et al., 2012; Simard, Yurovsky et al., 2009; Zweckberger et al., 2014). Consequently, SUR1–TRPM4 inhibitors have entered clinical trials for ischaemia-induced cerebral oedema (Sheth et al., 2018; Thompson et al., 2018; Zafardoost et al., 2016); however, most of these trials either did not achieve significant differences in the primary endpoints or were stopped early.

Astrocytes also regulate brain water balance and transport of water into and out of the brain parenchyma through the water channel proteins called aquaporins (AQPs). AQPs are historically known to be passive trans-porters of water. Lines of evidence in the last decade have highlighted the diverse function of AQPs beyond water homeostasis and that AQP water channels also facilitate transmembrane diffusion of small, polar solutes including mannitol and glycerol, which are used in treating TBI-associated oedema (Kitchen et al., 2015; Kitchen et al., 2019). Aquaporin-4 (AQP4) is the AQP mainly expressed in the CNS, is predominantly localised to astrocyte endfeet at the vasculature, and was historically considered the main channel that controls water flux between the blood and the brain (Badaut et al., 2014; Salman, Kitchen et al., 2021). Recent studies (Mestre et al., 2020) argue that since brain endothelial cells do not express any aquaporin channels, the endothelial membrane rather than the astrocyte membrane would be rate-limiting for water transport across the BBB, suggesting that perhaps AQP4 controls water flux between CSF in perivascular spaces and the brain parenchyma (Salman, Kitchen, Iliff & Bill, 2021), whereas water trans-port between blood and CSF is controlled by aquaporin-1 (Oshio et al., 2005) and choroid plexus transporters that co-transport water molecules (MacAulay, 2021).

AQP4 dysregulation has been reported in murine models of traumatic and ischaemic brain injury and is thought to have both detrimental and beneficial effects, facilitating the formation of cytotoxic oedema, but also facilitating the clearance of vasogenic oedema (Ren et al., 2013; Steiner et al., 2012). The expression of AQP4 has been reported to change following various brain insults and conflicting findings have been reported (Michinaga & Koyama 2015a). While some TBI rodent studies have reported elevated AQP4 expression (Ding et al 2009; Higashida et al., 2011; Kitchen et al 2020; Sun, Honey, Berk, Wong & Tsui, 2003; Tomura et al., 2011), others have reported reduced expression (Ke et al., 2001; Kiening et al., 2002; Zhao et al., 2005). In some cases, more careful immunohistochemical analysis demonstrates that AQP4 can be elevated in some regions (e.g. injury penumbra) whilst decreased in others (e.g. injury core) (Zhao et al., 2005). Increased AQP4 expression was reported in human tissue and cerebrospinal fluid following severe TBI (Hu et al., 2005; Lo Pizzo et al., 2013). Following a stroke, the expression of AQP4 was significantly increased between 6 and 48 h, which may aggravate the development of cyto-toxic oedema (Yu et al., 2015). Another study described a biphasic response, with expression reduced at 24 h in the infarct zone compared to control tissue but increased at 72 h (Zeng et al., 2012). Studies using AQP4-null mice demonstrated a protective effect of AQP4 deletion during the acute phase of stroke or TBI (Manley et al., 2000; Markou et al., 2022; Yao et al., 2015). Using global or astrocyte conditional (GFAP promoter) AQP4-knockout mice, AQP4 gene deletion was associated with a 35% and 31% decrease in cytotoxic oedema following stroke and systemic hypo-osmotic stress, respectively (Haj-Yasein et al., 2011; Manley et al., 2000). Interestingly, studies conducted during the later recovery phase have emphasised the role of AQP4 in oedema resolution. For example, the expression of AQP4 3 and 7 days following TBI in rats was inversely correlated with oedema, as quantified by T2-weighted MRI (Fukuda et al., 2012). Another study demonstrated that both upregulation of AQP4 by approximately six-fold and PLGB formation, 7−20 days after the induction of brain focal inflammation using lysolecithin in rats, was coincident with oedema resolution measured by *in vivo* MRI, whereas only a moderate threefold upregulation was reported during the acute oedema build-up phase (Tourdias et al., 2011). These results highlight the importance of astrocyte water channels for oedema resolution and restoration of water homeostasis. In the context of regeneration, it is inter-esting to note that $Aqp4^{-/-}$ mice have deficits in synaptic plasticity and spatial memory formation, indicating a possible role for AQP4 in the formation and tuning of the new synapses that are generated during recovery from stroke or TBI (Scharfman & Binder, 2013).

Although little attention has been paid to the Notch signalling pathway, it is essential for reactive astrogliosis and PLGB formation, and involved in brain recovery following injury. Administration of selegiline, a

monoamine oxidase inhibitor, between 2 and 9 days following focal ischaemic injury in rats increased mRNA and protein levels of the Notch-1 receptor and the Notch ligand Jagged-1 in peri-infarct astrocytes, significantly reduced peri-lesional oedema and increased the capillary network density (Nardai et al., 2015). The opposite effect was reported in a TBI model in which inhibition of gamma secretase, which is required for the proteolytic release of the Notch intracellular domain following receptor activation, mitigated vasogenic oedema, reduced reactive oxygen species generation and improved neurological function following TBI (Zhang, Chen, Liu & Zhao, 2018). However, given the large number of gamma secretase substrates, it is not entirely clear whether this protective effect was Notch-dependent. These findings were observed just 24 h after injury, so they are not directly comparable to the stroke model.

There is a large body of evidence supporting the involvement of astrocytes in modulating the inflammatory reaction following CNS injury (Giovannoni & Quintana 2020; Linnerbauer et al., 2020). Astrocytes respond to CNS insults by producing a variety of inflammatory mediators that vary with the type of injury and the time since injury. The activation of astrocytes within minutes following brain injury is associated with the release of proinflammatory cytokines and chemokines, such as tumour necrosis factor $\alpha$ (TNF$\alpha$), interleukin (IL)-1$\beta$, IL-6, IL-17, chemokine (C-X-C motif) ligand 1 (CXCL1) and CXCL9 and the activation of inflammatory TFs such as nuclear factor-$\kappa$B (NF-$\kappa$B) (Brambilla et al 2005; Hamby et al., 2012; John et al., 2005; Zamanian et al., 2012), which in turn exacerbate neuronal loss and functional impairment. However, many studies have described the beneficial roles of astrocytes in reducing brain damage and enhancing tissue repair after brain injury by segregating the injured area and limiting the spread of inflammation to adjacent viable tissue (Burda & Sofroniew, 2014). An early study found that, following forebrain injury, genetic ablation of newly proliferative reactive astrocytes and the associated disruption of PLGB formation aggravated and potentiated the inflammatory response and subsequent neuronal degeneration (Bush et al., 1999). Interestingly, these effects were maintained beyond the acute phase of the injury, with the ablation of reactive astrocytes accompanied by a 25-fold increase in the density of CD45-positive leukocytes (monocytes, neutrophils and lymphocytes) compared to controls 35 days after the injury. Likewise, a study in transgenic mice lacking the genes encoding the astrocyte intermediate filament proteins GFAP and vimentin reported similar findings following ischaemic injury with the loss of PLGB-forming astrocytes resulting in a spread of inflammation and greater infarct volume in the transgenic mice compared to their wild-type littermates 7 days after MCAO (Li et al., 2008a).

Transgenic mouse studies have demonstrated further mechanisms by which astrocytes protect against the expansion of inflammation beyond the lesioned area following brain injury. The expression of transforming factor $\beta$ (TGF-$\beta$), an inflammatory cytokine, and its signalling mediators is increased in astrocytes following ischaemic and traumatic brain injuries (Knuckey et al 1996; Rimaniol et al., 1995). Ast-Tbr2DN transgenic mice, which express a dominant negative inactive TGF-$\beta$ receptor specifically in astrocytes, had worse motor outcomes and increased density and activation of immune cells, without disturbing the physical astrocytic barrier following photothrombotic stroke in the motor cortex (Cekanaviciute et al., 2014). While there was no change in the infarct size at an early time point (day 1), it was significantly increased at 4 weeks, consistent with the reduced motor function reported at 3 and 4 weeks. These results suggest an important anti-inflammatory function of astrocytes through the immunoregulatory role of TGF-$\beta$ signalling, specifically in the late chronic phase of ischaemic injury. An anti-inflammatory role of reactive astrocytes in the late phase of injury is also suggested by studies on signal transducer and activator of transcription 3 (STAT3) in astrocytes and its requirement for PLGB formation. *Stat3* gene knockout in astrocytes resulted in the failure of astrocytes to proliferate and form a PLGB following moderate traumatic injury. These effects were associated with greater lesion volume, demyelination, the spread of inflammatory cells and attenuated motor recovery up to 28 days following the injury (Herrmann et al., 2008). Another study demonstrated that conditional knockout of STAT3 was associated with attenuated migration of astrocytes, extensive inflammatory cell infiltration and reduced motor function even at 6 weeks after spinal cord injury in mice (Okada et al., 2006). STAT3 was also reported to be required for the late-stage parallel orientation of elongated astrocyte processes and bundle formation near the lesion, a distinguishing feature of a mature PLGB following focal traumatic injury (Wanner et al., 2013). Moreover, in contrast to wild-type mice, STAT3 knockout mice exhibited a negative correlation between the density of CD-45 positive inflammatory cells and neuronal viability at 1−2 mm from the injury up to 21 days post-injury. In addition to its roles in oedema, Notch-1 signalling in astrocytes is required for reactive proliferating astrocytes to suppress the invasion of immune cells in the peri-infarct tissue following stroke (Shimada et al., 2011). However, this effect was only studied during the acute phase, 3 days after injury. Collectively, these findings provide convincing evidence that reactive astrocytes contribute to the recovery process following ischaemic and traumatic injury by inhibiting the invasion of immune cells and restricting the spread of injury to adjacent viable tissue.

## Astrocytes in angiogenesis and BBB repair

The BBB is a physiological barrier that regulates the transport of molecules into and out of the brain parenchyma. Astrocytes, specifically their endfeet that enwrap cerebral blood vessels, are a major cellular component of the BBB, alongside endothelial cells and pericytes. Disruption of BBB integrity occurs following ischaemic and traumatic brain injuries, and can happen directly as a result of the injury or indirectly as a result of inflammatory reactions and astrocyte dysfunction, and can itself lead to secondary injury due to increased permeability and the resultant vasogenic oedema and inflammatory damage (Hay et al 2015; Hoffmann et al., 2018). Thus, BBB repair has been posited as a therapeutic strategy following CNS injury, with a focus on astrocyte function. Several studies have suggested a dual role of astrocytes, with both beneficial and detrimental effects, in maintaining and/or disrupting BBB function after various types of brain injury (Michinaga & Koyama, 2019). This duality seems to arise from different distinct functions of astrocyte-derived molecules that control several pathophysiological processes such as leukocyte infiltration, neurogenesis, angiogenesis and neuroplasticity. An early study reported that BBB repair following penetrating TBI is dependent on reactive astrocytes specifically at late time points. Using transgenic mice expressing herpes simplex virus 1 thymidine kinase (HSV-TK) from the *Gfap* promoter, it was demonstrated that ablation of reactive astrocytes by treatment with ganciclovir reduced BBB repair at 14, 21 and 35 days after injury compared to controls, and that replacement of the transgenic astrocytes using control astrocyte grafts restored BBB integrity (Bush et al., 1999). A further study using an *in vitro* BBB model reported that astrocytes isolated from GFAP-deficient mice were impaired in their ability to induce barrier properties in co-cultured endothelial cells (measured by transendothelial transport of sucrose and 8-sulphophenyl-theophylline) (Pekny et al., 1998). Angiopoietin-1 (ANG-1), produced by endothelial cells and astrocytes, has been shown to have a protective effect on BBB function. Studies on ischaemic stroke rat models demonstrated that overexpression of ANG-1 reduced BBB leakage, lesion volume and enhanced functional recovery (Meng et al., 2014; Venkat et al., 2018). Moreover, ANG-1 overexpression stimulated an increase in vascular density and enhanced both angiogenesis and neurogenesis (via increased Neural progenitor cells (NPC) proliferation) 7 days after cerebral ischaemia. Similarly, pharmacological inhibition of the ANG-1 receptor Tie2, using a soluble Tie2–Fc fusion protein, aggravated BBB disruption and injury volume, and reduced vascular density and expression of barrier-associated genes such as *Ocln* (encoding occludin) and *Cdh5* (encoding VE-cadherin) following TBI in mice (Brickler et al., 2018). Inter-estingly, this study reported that a key feature in the ANG-1 protective signalling pathway was the activation of astrocytes, which was persistent even at 35 days after injury with high colocalisation of GFAP with CD31[+] vessels. Sonic hedgehog (SHH), a glycoprotein essential for angiogenesis and BBB formation and integrity (He et al., 2013; Tian & Kyriakides, 2009) is released by astrocytes and has been shown to have a beneficial effect in injury models at least partly through increasing the expression of ANG-1. A study in rats with permanent cerebral ischaemia demonstrated that intra-cerebroventricular injection of SHH led to reduced BBB disruption and oedema, and increased the expression of ANG-1 and the endothelial tight junctional proteins zonula occludens-1 (ZO-1) and occludin (OCLN) 7 days after ischaemic injury (Xia et al., 2013). These effects were observed *in vivo* and *in vitro* and were abolished by administrating either SHH or ANG-1 neutralising antibodies. Furthermore, a large body of evidence suggests that astrocyte-derived SHH increases the expression of various tight junction proteins and protects endothelial cells from apoptosis (Wang, Jin et al., 2014c; Zhu et al., 2015). SHH also reduced the levels of proinflammatory mediators, expression of cell adhesion molecules in endothelial cells and T cells, and leukocyte infiltration, *in vivo* and *in vitro*, suggesting an anti-inflammatory effect of astrocyte-derived SHH (Alvarez et al., 2011). Similar results were reported in a rat TBI model – SHH expression was increased approximately 1.5-fold at 28 days after TBI, and the administration of exogenous SHH alleviated BBB disruption, brain oedema, neuronal apoptosis, and inflammatory and oxidative damage, and improved neurological outcome (Wu et al., 2020).

Astrocyte-derived neurotrophic factors have also been shown to protect against BBB disruption. Glial-derived neurotrophic factor (GDNF) is important for postnatal development of the BBB and promotes angiogenesis (Chen et al., 2018). GDNF treatment increased the expression of endothelial tight junctional proteins (claudin-5, occludin and ZO-1), increased neuronal survival and improved endothelial cell barrier function (Igarashi et al., 2000; Shimizu et al., 2012; Xiao et al., 2014). Moreover, it was reported that mesencephalic astrocyte-derived neurotrophic factor (MANF) was neuroprotective in ischaemic and traumatic brain injury and attenuated BBB disruption in focal cerebral ischaemia and the Feeney free falling rat model (Li et al., 2018; Yu et al., 2010).

Vascular endothelial growth factor (VEGF) is a potent trophic factor that induces angiogenesis, regulates BBB integrity and is upregulated in the CNS following ischaemic and traumatic brain injuries (Dore-Duffy et al., 2007; Nag et al., 1997; Sun, Jin et al., 2003). Although VEGF increased BBB permeability in the acute stage following ischaemic brain injury (Wu et al.,

2018), late administration of exogenous VEGF promoted angiogenesis and improved neurological function without increasing BBB leakage at any time point up to 28 days after ischaemia (Zhang et al., 2000). Likewise, a study using a mouse model of TBI demonstrated that treatment with VEGF significantly increased neurogenesis and angiogenesis, and reduced lesion size 90 days after injury (Thau-Zuchman et al., 2010). These findings indicate that VEGF can be either damaging or protective, depending on the phase of the injury.

Like VEGF, matrix metalloproteinases (MMPs) have also been reported to elicit dual effects following CNS injury. MMPs are essential for angiogenesis and BBB integrity, and their expression is increased in reactive astrocytes after ischaemic and traumatic brain injury (Cunningham et al 2005; Hirose et al., 2013; Michinaga et al., 2018). Several studies have demonstrated that MMP-9 and MMP-2 overexpression during the early phase of ischaemic or haemorrhagic stroke leads to disruption of the BBB and degradation of tight junction proteins, enhancement of endothelial cell apoptosis and promotion of inflammatory cell migration (Min et al., 2015; Yang et al., 2007; Zhang, An, Wang, Gao & Zhou, 2018). Moreover, acute inhibition of MMP-9 attenuated BBB disruption in ischaemia–reperfusion injury and TBI rodent models (Michinaga et al 2018; Yang & Rosenberg, 2011). However, other studies have reported that although acute inhibition of MMPs attenuated BBB leakage, it also impaired long-term functional recovery following ischaemic stroke in rats (Sood et al., 2008). Furthermore, astrocyte-derived MMP-9 is beneficial during stroke recovery and is essential for ECM remodelling in the neurovascular unit (Rosell & Lo, 2008). It has been suggested that MMP-9, MMP-7 and MMP-2 contribute to angiogenesis, neurogenesis and vasculogenesis by digesting old ECM and increasing the availability of several growth factors, such as VEGF and BDNF growth factor (Rempe et al., 2016). MMP-9 expression was reported to be increased in the peri-infarct area following focal ischaemic stroke in rats and was colocalised with astrocytes and neurons 1−2 weeks after the injury. Delayed inhibition of MMPs at 1-week post-stroke exacerbated ischaemic brain injury and impaired functional recovery (Zhao et al., 2006). Together these findings highlight the contribution of MMPs to the late recovery phase following brain injury.

## Astrocytes in dendrogenesis, axogenesis and synaptogenesis

Neuronal maintenance, neurite outgrowth and repair of the neuronal network are coordinated by astrocytes (Araque et al 1999; Benarroch, 2005; Liu & Chopp, 2016).

During brain development and after ischaemic injury, astrogliogenesis occurs prior to neuronal maturation. The dynamic process of dendrogenesis and the formation of synapses then occurs in parallel with astrocyte maturation and morphogenesis (Freeman, 2010).

In functional synapses, presynaptic axons release neurotransmitters into the synaptic cleft, which activate receptors on the postsynaptic dendrite, leading to a neuronal signal (Hering & Sheng, 2001). Astrocyte processes physically envelop synapses and functionally interact with dendritic spines and synaptic terminals by releasing transmitters, such as ATP, adenosine and D-serine, to regulate synaptic function (Allen, 2014). In addition to contacting neurons, astrocytes are connected to each other by gap junctions, allowing nutrient molecules and ions to diffuse through an astrocytic syncytium, further expanding the range of synaptic regulation of neurons by astrocytes (Farhy-Tselnicker & Allen, 2018). The interaction between neurons and astrocytes modifies post-injury neuronal development via secreted factors and physical interactions (Withers et al., 2017). Astrogliosis occurs in early phases after ischaemic injury. This is characterised by the upregulation of GFAP, vimentin and chondroitin sulphate proteoglycans, and morphological hypertrophy (Yu et al., 2012). Astrogliosis inhibits axonal growth and creates a PLGB forming a physical and biochemical barrier between affected and unaffected regions. The barrier formed by reactive astrogliosis can also be detrimental as newly differentiated neurons recruited from neurogenic regions are likely to contribute to functional recovery following stroke, but may be unable to penetrate the PLGB (Arvidsson et al., 2002).

Neurogenesis and repair are modulated by astrocytes that regulate dendritic growth and synaptogenesis via paracrine and contact-mediated signals. Dendrogenesis is guided to orient new dendrites toward nearby astrocytes, but dendrite growth and neurite outgrowth (Maldonado et al., 2017) are locally restricted upon physical interaction with astrocytes, resulting in significant asymmetry in the neuron's dendritic arbour (Withers et al., 2017). Crucially, this effect was dependent on engagement of astrocyte $\alpha v \beta 3$ integrin with neuronal Thy-1, suggesting that targeting of this interaction could be a useful strategy to modulate the effect of astrocytes on dendrite outgrowth (Maldonado et al., 2017). Furthermore, $\alpha v \beta 3$ integrin was reported to mediate changes in astrocyte morphology and adhesion to ECM components, which are significantly altered in reactive astrocytes and may inhibit repair capacity (Pérez et al., 2021). $\beta 1$ integrin in reactive astrocytes was reported to facilitate upregulation of N-cadherin, engagement of astrocytes with lesional collagen-I, and transformation of reactive astrocytes into 'scar-forming' astrocytes following spinal cord injury. This was inhibited by a $\beta 1$ integrin blocking antibody, again suggesting that targeting of integrin signalling could be a

useful therapeutic strategy to promote repair (Pérez et al., 2021).

Astrocytes may promote neurite regeneration by producing fibronectin (Tom et al., 2004), while secreted thrombospondins and high endothelial venule protein (HEVIN) increase the formation of structurally normal but silent synapses. Secreted protein acidic and rich in cysteine (SPARC) antagonises the synaptogenic function of HEVIN, giving astrocytes control over local excitatory synaptogenesis (Kucukdereli et al., 2011). Several studies reported that thrombospondins are upregulated for up to 14 days following focal ischaemic injury in mice and were mostly colocalised with astrocyte markers (Christopherson et al., 2005; Liauw et al., 2008). In this context, thrombospondins knockout in mice leads to impaired axonal sprouting, synaptic density and as a result impaired recovery after stroke, supporting the importance of astrocyte-derived thrombospondins in mediating cortical plasticity and recovery following brain injury. In parallel with this evidence, astrocyte-secreted glypicans 4 and 6 (Gpc4 and Gpc6) were reported to activate these silent synapses and increase the frequency and amplitude of glutamatergic synaptic events by increasing the surface localisation and clustering of receptors (Allen et al., 2012).

Exosomes, a type of small extracellular vesicle (EV) (Théry et al., 2018), mediate regulatory cell-to-cell information exchange between glia and neurons. Exosomes are membrane vesicles of endocytic origin that differ from other EVs in their size, density and lipid, protein and nucleic acid composition (Gould & Raposo, 2013) and facilitate intercellular communication to regulate recipient cell function (Robbins & Morelli, 2014). Exosomes are transported into cells either by direct fusion with the plasma membrane or by internalisation following binding to cell surface proteins, such as adhesion molecules (Colombo et al., 2014). Deng et al. (2018) reported that astrocyte-derived small EVs increased neuronal phosphorylation of AMP-activated protein kinase, and the activity of phosphoinositide 3-kinase, p85, Akt and mechanistic target of rapamycin post-injury, normalised inflammation by decreasing the expression of cyclooxygenase-2, inducible nitric oxide synthase and TNF$\alpha$ and reduced apoptotic markers including caspase-3, Bax and Bcl-2. Astrocytes have also been reported to affect dendritic arborisation and axonal growth by modifying the miRNA cargo of small EVs delivered to neurons (Luarte et al., 2020).

The structural interactions of astrocytes with neurons facilitate activity-dependent plasticity in the mature CNS, which may contribute to learning and memory (Perez-Alvarez et al., 2014). Functional neural connectivity requires precise synaptic remodelling and active scavenging of unnecessary synapses. Astrocytes play a major role in synaptogenesis and synapse maintenance and mediate postnatal synapse elimination by phagocytic activity (Chung et al 2013). Davis et al. (2014) observed the sequestration of degraded axonal mitochondria by local astrocytes. More recently, Hayakawa et al. (2016) demonstrated that astrocytes are capable of direct transfer of functional mitochondria to neurons, and that suppression of this process worsens ischaemic injury. Synthesis and proteolysis of specific ECM elements in specialised domains known as peri-neuronal nets (PNN) play a critical role in synaptic plasticity and neuronal activity (Bosiacki et al., 2019). The disruption of PNNs has been observed following a variety of brain injuries and disorders, highlighting their importance in maintaining CNS physiological functions (Hsieh et al., 2017; Kim et al., 2016; Patel et al., 2019). Astrocytes, as a major source of PNN components including hyaluronic acid, CSPGs and tenascins contribute significantly to PNN remodelling (Tewari et al., 2022). Together, these studies support the involvement of astrocytes in regulating neuroplasticity and modulating neuronal activity in physiological and disease states.

### Astrocytes in neurogenesis

Neurogenesis is a set of processes that take place during development, and in the mature brain in response to various stimuli including seizure, ischaemia and physical exercise. It involves the formation of new neurons from neural stem cells (Kokaia & Lindvall 2003; Lledo et al., 2006; Parent et al., 2006). In adult brain, two regions have been identified as neurogenic: the hippocampal dentate gyrus (subgranular zone) and the lateral ventricle (SVZ) (Ming & Song, 2005). Astrocytes are one of the major cellular constituents of the neurogenic niches in these regions and participate directly in the proliferation, migration, differentiation and synaptic integration of newborn neurons through membrane-associated molecules and indirectly by secreting numerous neurogenesis-promoting factors (Cassé et al., 2018; Shetty et al 2005; Song et al., 2002). Furthermore, mature astrocytes could be a direct cell source for neurogenesis as they possess intrinsic neuro-genic potential and can be reprogrammed to neurons through dedifferentiation into multipotent neurosphere cells, a process highly activated during brain injury (Magnusson et al., 2014; Robel et al., 2011).

The release of astrocyte-derived factors plays a vital role in promoting various stages of neural stem cell maturation, and the interplay between mature astrocytes and neurons is essential for establishing functional neuronal circuitry. Blocking vesicle release by astrocytes reduced survival of new-born neurons, inhibited dendritic maturation and reduced branching and density of the dendritic spines leading to impairment of glutamatergic synaptic input and functional integration

of adult-born neurons in mice (Sultan et al., 2015). BDNF is one of the main trophic factors released by astrocytes in the hippocampus and contributes to the maturation of neurons and synaptic plasticity. In cortical neurons, BDNF binds to TrkB and TrkC receptors and promotes neuroplasticity and the growth of new-born neurons (Bergami et al., 2008). Puehringer et al. (2013) showed that in newborn neurons astrocytic epidermal growth factor (EGF) activated neuronal EGF receptor leading to membrane translocation of neural TrkB and TrkC receptors, promoting signal responsiveness of the newborn neurons. Additionally, EGF induces neuronal differentiation, migration and proliferation of stem cells in the CNS (Araujo et al., 2019). D-Serine is another factor secreted by astrocytes during the recovery process and has been shown to be essential for the synaptic integration of newborn neurons in the pre-existing neuronal circuit (Sultan et al., 2015). Usually, L-serine is synthesised in astrocytes and converted to D-serine by serine racemase, a pyridoxal phosphate-dependent enzyme that is expressed in neurons and astrocytes (Wolosker & Balu, 2020). D-Serine serves as the endogenous NMDA receptor co-agonist and participates in hippocampal long-term potentiation (Henneberger et al., 2010). Furthermore, it promotes hippocampal neurogenesis by increasing the proliferation, differentiation and density of neural stem cells as well as by improving the survival of newborn neurons (Huang et al., 2012; Sultan et al., 2013). A recent preprint by Roychaudhuri et al. (2023) reported that serine racemase regulates adult neurogenesis in the SVZ by controlling *de novo* fatty acid synthesis and that the deletion of serine racemase in nestin precursor cells diminished SVZ neurogenesis, which could be rescued by administration of L- and D-serine.

Astrocyte reprogramming has attracted much attention recently and is considered a promising strategy for replacing dead neurons following brain injury (Jiang et al., 2021). Astrocytes and neurons originate from the same progenitor cells, and mounting evidence suggests the feasibility of astrocyte reprogramming into neurons using specific TFs or microRNAs (Peng et al., 2022). Expression of the TF NeuroD1 in mouse brain following stroke using an AAV reprogrammed astrocytes to neurons, leading to improvement in motor and cognitive function (Chen, Ma et al., 2020). The same group also replicated this study in macaques (Ge et al., 2020). Another study reported the conversion of astrocytes into neuronal progenitor cells by overexpression of *SOX2*, followed by differentiation into mature neurons by treatment with neurotrophic factors in a mouse spinal cord injury model (Su et al., 2014). miR-365 knockout was reported to induce the conversion of astrocytes into neurons by up-regulating *Pax6* (Mo et al., 2018). Treatment with miR-124 activated neuronal differentiation of a mouse cancer cell line by targeting polypyrimidine tract binding protein 1 (PTB1), which in

turn increased the expression of a splice variant of PTB2, important for correct splicing of several neuronal genes (Makeyev et al., 2007).

Astrocytes are active players in the process of neurogenesis and can be reprogrammed directly into neurons or release substances that target neurogenic niches to promote neural stem cell proliferation and differentiation, and integration of newly differentiated neurons into mature neuronal circuits.

## Potential therapeutic targets to enhance functional recovery

Astrocytes are involved in recovery, and their extensively branched processes constitute multiple contact points with each other and with the BBB, synapses and neurons, which are all essential factors in the recovery process (Chiareli et al., 2021). Astrocytes actively participate in neurotransmitter uptake and release, providing neurons with a recovery-permissive environment, synthesising and releasing neurotrophic factors, replacing damaged neurons by transdifferentiation, and may participate in other mechanisms that are yet to be discovered. All of these mechanisms have been successfully manipulated by specifically targeting astrocytes and have shown promising recovery-promoting effects in various rodent models of stroke, TBI and spinal cord injury (SCI), which are tabulated in Table 1. All studies included in the table are *in vivo* and measured motor and/or sensory functions at least 2 weeks after injury, which is a good indication of functional recovery.

During the recovery phase of injury, accumulation of the inhibitory neurotransmitter GABA causes hypo-excitability of neurons in both the perilesional area and areas that are functionally connected to it, impeding functional recovery (Brickley & Mody, 2012). Astrocytes express transporter proteins for excitatory and inhibitory neurotransmitters and promote functional recovery by re-establishing the excitation/inhibition balance via glutamate (GLT-1) and GABA (GAT) transporters, respectively. Targeting astrocytic GABA handling could therefore be a useful therapeutic strategy.

It has long been debated whether the PLGB has an overall positive or negative effect on neurorecovery, and there is evidence to support both viewpoints depending on the interpretation of data (Anderson et al., 2016; Liddelow & Barres, 2016). The PLGB forms a dense barrier preventing axonal regeneration while at the same time limiting the lesioned area and inhibiting the spread of injury to healthy tissue (Cregg et al., 2014; Kawano et al., 2012). Indeed, Gu et al., 2019 have recently shown that conditional ablation of reactive astrocytes exacerbates neuroinflammation and hampers spontaneous functional recovery mainly by interfering with PLGB formation (Gu et al., 2019). On the other hand, decreasing PLGB

**Table 1. Potential astrocyte-related targets involved in functional recovery**

| Type of target | Intervention and significance | Model | Reference |
|---|---|---|---|
| Gliotransmitter uptake, synthesis and release | Inhibition of GABA synthesis by selective and reversible inhibitor of monoamine oxidase-B KDS2010, reduced cortical diaschisis and improved motor function assessed by single-pellet reaching task and cylinder test | Photothrombotic capsular infarction in rats | Nam et al. (2020b) |
| | Administration of SNAP-5114 which is selective antagonist for astrocytic GABA transporter GAT- 3/4 reduced the foot faults and asymmetry index and improved functional recovery | Photothrombotic stroke model in mice | Lin et al. (2018) |
| | Injection of an adeno-associated viral (AAV) vector expressing the rat GLT-1 cDNA (AAV-GLT1) prior to infarction significantly reduced body asymmetry in an elevated body swing test and neurological abnormality scores in Bederson's test 2 weeks following ischaemia | Transient middle cerebral artery occlusion (tMCAO) model of ischaemia in rats | Harvey et al. (2011) |
| | Activation of astrocytic GLT-1 with ipsilateral limb stimulation enhanced the somatosensory function while inhibition of astrocytic GLT-1 with TFB-TBOA disturbed the somatosensory function | Photothrombotic stroke model in mice | Takatsuru et al. (2013) |
| Reactive astrogliosis, scar and the associated mediators (extracellular matrix, growth cone inhibitors) | Selective ablation of the scar by the herpes simplex virus thymidine kinase/ganciclovir (HSV-TK/GCV) system failed to improve spontaneous functional recovery | Spinal cord injury in mice | Gu et al. (2019) |
| | Genetic deletion of astrocyte EphA4 decreased reactive astrogliosis and scarring and improves axonal sprouting and neurite outgrowth | Spinal cord injury in mice | Goldshmit et al. (2004) |
| | Perilesional infusion of chondroitinase ABC 7 days after stroke increased BDNF expression and neurite outgrowth with noticeable improvement in functional recovery | Permanent MCAO in rats | Hill et al. (2012) |
| | Inhibition of Ephrin-A5 with a soluble receptor decoy 7 days after stroke improved axonal sprouting in motor, premotor and prefrontal circuits and enhanced functional recovery | Photothrombotic stroke model in mice | Overman et al. (2012) |
| Neurotrophic factors | Delivery of mesencephalic astrocyte-derived neurotrophic factor to the peri-infarct area (MANF) enhanced the reversal of behavioural deficits | Distal MCAO model of ischaemia–reperfusion injury in rats | Mätlik et al. (2018) |
| | Administration of galectin-1 increased BDNF levels and improved neurological severity score and functional recovery | Photothrombotic stroke model in rats | Qu et al. (2010) |
| Transcription factors | Conditional knockout of Socs3 (a negative regulator of *Stat3*) activated *Stat3* and improved motor function | Spinal cord injury in mice | Okada et al. (2006) |
| | Reprogramming of astrocytes into neurons by lentivirus-mediated expression of NeuroD1 reduced astrogliosis and improved locomotor, sensorimotor and psychological functions | Permanent right MCAO in mice | Jiang et al. (2021) |

*(Continued)*

**Table 1. (Continued)**

| Type of target | Intervention and significance | Model | Reference |
|---|---|---|---|
| | Administration of Tert-butylhydroquinone activated Nrf2 pathway and improved motor function and neurological deficits | Permanent MCAO in mice | Chen, Zhang et al. (2019a) |
| | Astrocyte specific knockout of *SOX2* reduced astrocytes activation and improved functional recovery | Controlled cortical impact (CCI) model of TBI in mice | Chen, Zhong et al. (2019a) |
| | Targeted knockdown of astrocytes Zeb2 impeded motor recovery | SCI and transient MCAO models in mice | Vivinetto et al. (2020) |
| | Conditional knockout of ApoE from astrocytes demonstrated impairment in dendritic tree complexity and in reversal of learning by Morris water maze | Controlled cortical impact (CCI) model of TBI in mice | Yu et al. (2021) |
| miRNAs | miR-365 knockout induced the conversion of astrocytes into neurons by up-regulating the 3′-UTR *Pax6* | Transient MCAO in adult rats | Mo et al. (2018) |
| Others | Genetic deletion or pharmacologic inhibition of astrocytes transglutaminase 2 with irreversible TG2 transglutaminase inhibitor VA4 improved locomotor function 42 days following spinal cord injury | Mouse spinal cord contusion injury | Elahi et al. (2021) |
| | Activation of sigma-1 receptor with the agonist SA4503 promoted neurite outgrowth and spinogenesis | Permanent and transient MCAO in rats | Ruscher et al. (2011) |
| | Inhibition of AQP4 localisation to the blood-spinal cord barrier by trifluoperazine reduced brain oedema and enhanced functional recovery | SCI model in rats | Kitchen et al. (2020) |
| | Overexpression of omega-3 polyunsaturated fatty acids in mice enhances angiogenesis and improves long-term functional recovery | Transient focal cerebral ischaemia in mice | Wang, Shi et al. (2014b) |

formation by genetic deletion of EphA4 improved axonal sprouting and neurite outgrowth (Goldshmit et al., 2004). Additionally, reactive astrocytes produce several growth inhibitory factors such as CSPG and Ephrin-A5 and inhibition of these factors with genetic, biochemical and pharmacological approaches promote neurite outgrowth and axonal sprouting and enhance functional recovery (Hill et al., 2012; Overman et al., 2012).

Trophic factors including nerve growth factor (NGF), GDNF, ciliary neurotrophic factor, BDNF, basic fibroblast growth factor (bFGF), VEGF and erythropoietin are released by reactive astrocytes following injury to protect the damaged neurons during the acute phase and contribute to neurorepair and plasticity during the recovery phase (Ridet et al., 1997). There is a growing body of evidence that such trophic factors influence neurogenesis, gliogenesis, synaptogenesis, axogenesis, neuritogenesis and angiogenesis, which altogether are indispensable for functional recovery (Liu & Chopp, 2016). Boosting astrocytic BDNF release by administration of galectin-1 led to significant improvement in functional recovery in an ischaemic rat model (Qu et al., 2010), and selegiline administration, proposed to increase NGF and FGF, led to modest improvement in one primary endpoint in a phase II stroke trial (Sivenius et al., 2001).

Although difficult to drug, TFs are attractive therapeutic targets since they integrate the cellular responses of multiple signalling pathways and therefore control many physiological and pathological processes. STAT3, cAMP response element binding protein (CREB) and NRF-2 are among the most important TFs that mediate the astrocyte response to injury and drive neuro-restoration. STAT3 is a master regulator of astrocyte activation and scar formation (Ceyzériat et al., 2016). In contusive SCI, astrocyte-specific deletion of *Stat3* impaired astrocyte migration and enhanced neuro-inflammation and demyelination leading to motor deficits. Furthermore, STAT3 regulates expression of the synaptogenic molecule thrombospondin-1 and facilitates the recovery of excitatory synapses onto axotomised motor neurons in adult mice (Tyzack et al., 2014). CREB

regulates the expression of BDNF, which in turn regulates its expression by activating CREB, forming a transcriptional positive feedback loop (Chiareli et al., 2021). Importantly, CREB has a vital role in neuroplasticity and long-term memory and controls the expression of the glutamate transporter genes *EAAT2* (encoding GLT1) and *EAAT1* (encoding glutamate-aspartate transporter (GLAST)) in rat astrocytes (Karki et al., 2013). However, the role of astrocytic CREB in functional recovery following brain injury is yet to be investigated. Targeting other astrocyte-specific TFs such as *SOX2*, Zeb2 and apoE

(Theendakara et al., 2018) influenced neurorecovery by interfering with various neurorestorative mechanisms (Chen, Zhong et al., 2019; Vivinetto et al 2020; Yu et al., 2021) (Table 1).

Recent studies have demonstrated the feasibility of reprogramming astrocytes into neurons either directly by expression of reprogramming genes or indirectly by astrocyte dedifferentiation into adult neural stem cells (Chiareli et al., 2021). In principle, this approach could compensate for lost neurons and restore impaired function by integrating new neurons

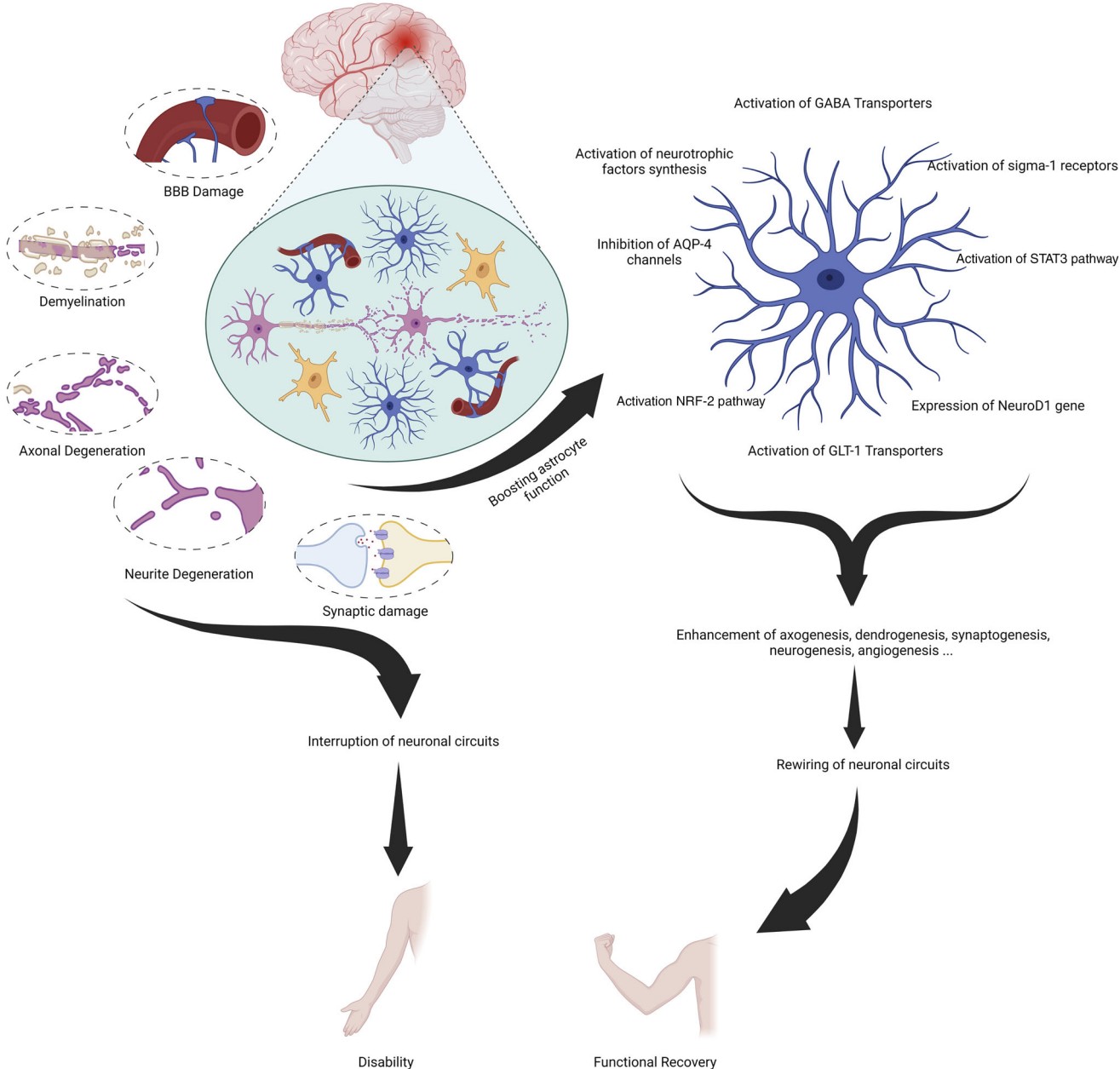

**Figure 2. Astrocytes-mediated recovery mechanisms**
This summary figure shows the common mechanisms of stroke-induced neurodegeneration and disability and the role of astrocytes in boosting the recovery process.

into the disconnected circuits. For example, a recent study showed that overexpression of the NeuroD1 gene in the peri-infarct area in mice was sufficient to reprogram reactive astrocytes into neurons and restore interrupted cortical circuits and synaptic plasticity (Table 1). Similarly, the expression of *SOX2* in astrocytes of adult and aged mouse brains reprogrammed them into proliferative neuroblasts, which after administration of BDNF and noggin, developed into functional neurons (Niu et al., 2013). Therefore, reprogramming astrocytes by targeting specific TFs has emerged as a novel avenue in neuro-restorative therapy.

Astrocytes have numerous targets, some of which have been successfully drugged to improve functional recovery in rodent models (Table 1). Astrocytes are attractive therapeutic targets, with the potential impact on multiple neurorecovery mechanisms to enhance functional recovery. As well as their roles in neurorecovery following injury, the neurorecovery and neurorestorative role of astrocytes in various neurodegenerative diseases have also been reviewed elsewhere (Chiareli et al 2021; Liu & Chopp 2016; Sims & Yew, 2017).

## Conclusion

The recovery process following CNS injury is slow and often incomplete and involves intricate interaction between various cell types and signalling pathways (Fig. 2). Astrocytes provide structural and functional support to neurons, and interact with a variety of other cell types, making them crucial for successful functional recovery. Astrocytes can augment angiogenesis, neurogenesis, synaptogenesis, dendrogenesis and axogenesis and establish a recovery-permissive milieu. Following brain injury, astrocytes develop a morphologically and functionally distinct reactive phenotype, which facilitates the clearance of dead cell debris and produces trophic factors to repair damaged neurons and blood vessels. However, reactive astrocytes may also produce growth-inhibitory factors and form a PLGB, both of which inhibit the recovery process. As a result, astrocytes are a prospective therapeutic target for pharmacological therapies due to their impacts on functional outcome and neurological recovery.

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

## Additional information

### Competing interests

The authors declare no conflicts of interest.

### Author contributions

Q.M.A and G.A.B. wrote the first draft of the manuscript together with O.A., P.K., Z.A.S. and M.M.S. G.A.B. composed the figures. M.M.S. and P.K. planned, edited and supervised the writing and reviewed the manuscript. All authors have read and approved the final version of this manuscript and agree to be accountable for all aspects of the work in ensuring that questions related to the accuracy or integrity of any part of the work are appropriately investigated and resolved. All persons designated as authors qualify for authorship, and all those who qualify for authorship are listed.

### Funding

P.K. is supported by a Discovery fellowship from the UK Biotechnology and Biological Sciences Research Council (BB/W00934X/1). M.M.S is supported by the Medical Research Council (MRC) Career Development Award (MR/W027119/1).

### Keywords

astrocytes, brain injury, functional recovery, regeneration, traumatic CNS injury

## Supporting information

Additional supporting information can be found online in the Supporting Information section at the end of the HTML view of the article. Supporting information files available:

**Peer Review History**

