## [Peer Review History · The Journal of Physiology]

Astrocytes in functional recovery following CNS injuries

Qasim M. Alhadidi, Ghaith A. Bahader, Oiva Arvola, Philip Kitchen, Zahoor A. Shah, and Mootaz Salman
DOI: 10.1113/JP284197

Corresponding author(s): Mootaz Salman (mootaz.salman@dpag.ox.ac.uk)

The following individual(s) involved in review of this submission have agreed to reveal their identity: Alexej Verkhratsky (Referee #1)

Review Timeline:

Submission Date:	29-May-2023
Editorial Decision:	13-Jun-2023
Revision Received:	02-Aug-2023
Accepted:	07-Aug-2023

Senior Editor: Peking Fong

Reviewing Editor: Robert Fenton

Transaction Report:

Dear Dr Salman,

Re: JP-SR-2023-284197 "Emerging roles of astrocytes in functional recovery following brain injury" by Mootaz Salman, Qasim M. Alhadidi, Ghaith A. Bahader, Oiva Arvola, Philip Kitchen, and Zahoor A. Shah

Thank you for submitting your manuscript to The Journal of Physiology. It has been assessed by a Reviewing Editor and by 2 expert referees and we are pleased to tell you that it is acceptable for publication following satisfactory revision.

ABSTRACT FIGURES: Authors may use The Journal's premium BioRender account to create/redraw their Abstract Figures (and any other suitable schematic figures). Information on how to access this account is here: <https://physoc.onlinelibrary.wiley.com/journal/14697793/biorender-access>.

REVISION CHECKLIST: Upload a full Response to Referees file. To create your 'Response to Referees' copy all the reports, including any comments from the Senior and Reviewing Editors, into a Microsoft Word, or similar, file and respond to each point, using font or background colour to distinguish comments and responses and upload as the required file type.

- 'Potential Cover Art' for consideration as the issue's cover image.
- Appropriate Supporting Information (Video, audio or data set: see https://jp.msubmit.net/cgi-bin/main.plex?form_type=display_requirements#supp).

We look forward to receiving your revised submission.

Yours sincerely,

Dr Peying Fong
Senior Editor
The Journal of Physiology
<https://jp.msubmit.net>
<http://jp.physoc.org>
The Physiological Society
Hodgkin Huxley House
30 Farringdon Lane
London, EC1R 3AW
UK
<http://www.physoc.org>
<http://journals.physoc.org>

REQUIRED ITEMS

-Please include an Abstract Figure file, as well as the figure legend text within the main article file. The Abstract Figure is a piece of artwork designed to give readers an immediate understanding of the Review Article and should summarise the main conclusions. If possible, the image should be easily 'readable' from left to right or top to bottom. It should show the physiological relevance of the Review so readers can assess the importance and content of the article. Abstract Figures should not merely recapitulate other figures in the Review. Please try to keep the diagram as simple as possible and without superfluous information that may distract from the main conclusion of the Review. Abstract Figures must be provided by authors no later than the revised manuscript stage and should be uploaded as a separate file during online submission labelled as File Type 'Abstract Figure'. Please ensure that you include the figure legend in the main article file. All Abstract Figures will be sent to a professional illustrator for redrawing and you may be asked to approve the redrawn figure before your paper is accepted.

-Your MS must include a complete "Additional information section" with the following 4 headings and content:

Competing Interests: A statement regarding competing interests. If there are no competing interests, a statement to this effect must be included. All authors should disclose any conflict of interest in accordance with journal policy.

Author contributions: Each author should take responsibility for a particular section of the study and have contributed to writing the paper. Acquisition of funding, administrative support or the collection of data alone does not justify authorship; these contributions to the study should be listed in the Acknowledgements. Additional information such as 'X and Y have contributed equally to this work' may be added as a footnote on the title page.

It must be stated that all authors approved the final version of the manuscript and that all persons designated as authors qualify for authorship, and all those who qualify for authorship are listed.

Funding: Authors must indicate all sources of funding, including grant numbers. If authors have not received funding, this must be stated.

It is the responsibility of authors funded by RCUK to adhere to their policy regarding funding sources and underlying research material. The policy requires funding information to be included within the acknowledgement section of a paper. Guidance on how to acknowledge funding information is provided by the Research Information Network. The policy also requires all research papers, if applicable, to include a statement on how any underlying research materials, such as data, samples or models, can be accessed. However, the policy does not require that the data must be made open. If there are considered to be good or compelling reasons to protect access to the data, for example commercial confidentiality or legitimate sensitivities around data derived from potentially identifiable human participants, these should be included in the statement.

Acknowledgements: Acknowledgements should be the minimum consistent with courtesy. The wording of acknowledgements of scientific assistance or advice must have been seen and approved by the persons concerned. This section should not include details of funding.

-The Reference List must be in Journal format https://jp.msubmit.net/cgi-bin/main.plex?form_type=display_requirements#refs

-Please upload separate high quality figure files via the submission form.

-Author profile(s) must be uploaded via the submission form. Authors should submit a short biography (no more than 100 words for one author or 150 words in total for two authors) and a portrait photograph of the two leading authors on the paper. These should be uploaded, clearly labelled, with the manuscript submission. Any standard image format for the photograph is acceptable, but the resolution should be at least 300 dpi and preferably more. A group photograph of all authors is also acceptable, providing the biography for the whole group does not exceed 150 words.

EDITOR COMMENTS

Reviewing Editor:

All reviewers had valid comments to address - especially reviewer #2. These should be easily addressed with some additional rewriting and added information.

Senior Editor:

Your invited Symposium Review has been reviewed by two Expert Referees and a Reviewing Editor. Overall, they agree this is an important and underappreciated topic, and that a well-considered review carries potential to be impactful.

However, as you will see from the comments, they identify some critical points for your consideration as you revise your manuscript. Although Referee 1's first three points regarding terminology are straightforward and likely can be readily addressed, there nonetheless were concerns from both Referees regarding both breadth and depth of coverage. Referee 1's comments 4-7, and the bulk of feedback received from Referee 2, identify important gaps in content that will require your deeper consideration. The Reviewing Editor and I feel the points raised by both Referees are valid. Filling the identified conceptual and informational gaps will enrich content and increase reader engagement with this important topic. In this spirit, I encourage you to address all critiques fully.

REFEREE COMMENTS

Referee #1:

JP-SR-2023-284197

Emerging roles of astrocytes in functional recovery following brain injury

General comments: Timely and needed review highlighting the role of astrocytes in the regeneration of the CNS tissue after trauma. Overall the review is well written well structured and delivers the message. I support publication.

Minor comments:

1. Terminology: "activated astrocytes" as used by authors is incorrect. Astrocytes (as well as other glia) are constantly activated in physiological settings by various inputs. In the context of trauma astrocytes become REACTIVE (reactive astrogliosis; reactive astrocytes), see PMID 33589835.

2. Terminology: "glial scar" does not exist; all scars (in skin or in the brain) are of fibrotic nature; the scar is the replacement of parenchymal cells (which for the CNS are neurones and glia) with stromal cells (pericytes and fibroblasts) which produce matrix. Glial cells erect the perilesional barrier. (see PMID 34164732; doi: 10.1016/B978-0-12-821565-4.00005-5; doi 10.1016/B978-0-12-821565-4.00013-4).

3. Terminology: "The finely branched astrocytic processes contact all cellular components of the CNS and enable a single astrocyte to contact up to 100,000 neurons (16, 17)". Astrocytic processes are classified as branches (primary processes) and leaflets (terminal processes without organelles, which mainly contact synapses - PMID 34479758). In human brain a single protoplasmic astrocyte contacts ~ 2Mil synapses.

4. Probably it might help to the reader if the authors add a paragraph on astrogliopathy; pointing that besides reactivity, in many pathologies astrocytes undergo atrophy and loss of function; even reactive astrocytes can demonstrate either gain or loss of function. Downregulation of glutamate uptake for example is the leading mechanism of secondary neurotoxicity). For details see doi: 10.1016/B978-0-12-821565-4.00005-5; PMID 28805002).

5. A figure summarising major mechanisms that can be involved in boosting astrocytes function in order to enhance functional recovery after traumatic brain injuries.

6. Tonic GABA inhibition - mid that reactive astrocytes may overexpression MAO-B, hence increasing synthesis and secretion of GABA - look at multiple works of Justin Lee; see also PMID 30877782.

7. When discussing about Na⁺, may be some references to overall Na homeostasis and signalling in astrocytes can be if value - see PMID 30734296, PMID 26919326.

Referee #2:

The review is well written and describes fundamental roles of astrocytes. However, some sections of the review are very basic and does not add much to the existing reviews available on the topic. Some sections are not even astrocyte specific. For example, in the section on mechanisms of functional recovery, in the 5-6 pages of the text, there was no specific reference on astrocytes role, which is the topic of this review. Overall, some new information beyond what is described extensively in the past decades and in other reviews are needed to inform the readers about recent advances on astrocyte roles and mechanisms.

I have provided some comments to help the authors elevate the content of the review.

- Astrocyte heterogeneity has been an important topic in recent years. A section on this topic needs to be included in the manuscript. There are several recent single cell RNA sequencing and RNA in situ analysis from various research groups (e.g. studies by the Holt and Rowitch labs).
- In the synaptogenesis section, role of astrocytes in the formation of peri-neuronal net (PNN) should be discussed as it is highly relevant to neuronal activity and plasticity.

- For Thrombospondin, it is important to include the seminal work by Barres lab on astroglial production of this critical molecule in the synapse formation section.
- Integrin signaling has been shown to be important for the interaction between astrocytes and the ECM in development and after injury, as well as other cell types including endothelial cells. Some discussion on this family of proteins should be included.
- Discussion on CSPGs should be expanded as they play an important role in CNS repair. There have been major advances in targeting CSPGs more than chondroitinase (e.g. targeting their signaling receptor in axon regeneration, on neural and oligodendrocyte progenitor response after brain/spinal cord injury).
- GFAP in CSF after TBI/concussion is a good biomarker and predictor of functional outcomes. The role of GFAP as a predictive biomarker should be discussed.

A minor comment: There is a need for overall proof-reading throughout the manuscript for grammar and typos.

END OF COMMENTS

Confidential Review

29-May-2023

Professor Peying Fong, Associate Editor
Professor Peter Kohl, Editor-in-Chief
Journal of Physiology

29th July 2023

Dear Professor Fong,

Thank you for your e-mail of 13th of June inviting us to submit a revised version of manuscript (Review, No. JP-SR-2023-284197), now titled "*Astrocytes in functional recovery following CNS injuries*".

We thank you and the two referees for the careful scrutiny of our work. We were glad to see the two referees describing our review as "Timely and needed", "well written" and "describes fundamental roles of astrocytes".

In our revised manuscript, we have clarified and strengthened our review to directly address the substantive points around conceptual coverage and citation raised by you and the referees. Accordingly, we have edited our manuscript to incorporate:

- More than 20 lines have been deleted while their major scientific content has been added to the new restructured sections.
- One new figure (Figure 2) and updated Figure 1;
- More than 20 new and up-to-date references;

Our detailed responses are given below; the referees' comments are in italics and our responses are highlighted below each one in blue text. Only the major corresponding textual changes are also marked in blue text in our revised manuscript.

We trust that in revising our manuscript, we have restructured it and provided robust responses, evidenced by comprehensive referencing to the published literature, that will satisfy the concerns raised, and that our *review* will now be acceptable for publication in *Journal of Physiology*.

I look forward to hearing from you.

With my best wishes,

Dr Mootaz Salman (on behalf of all the co-authors)

Response to the Reviewing Editor:

All reviewers had valid comments to address - especially reviewer #2. These should be easily addressed with some additional rewriting and added information.

We have addressed all the reviewers' comments and added and revised the suggested topics.

Response to the Senior Editor:

Your invited Symposium Review has been reviewed by two Expert Referees and a Reviewing Editor. Overall, they agree this is an important and underappreciated topic, and that a well-considered review carries potential to be impactful.

However, as you will see from the comments, they identify some critical points for your consideration as you revise your manuscript. Although Referee 1's first three points regarding terminology are straightforward and likely can be readily addressed, there nonetheless were concerns from both Referees regarding both breadth and depth of coverage. Referee 1's comments 4-7, and the bulk of feedback received from Referee 2, identify important gaps in content that will require your deeper consideration. The Reviewing Editor and I feel the points raised by both Referees are valid. Filling the identified conceptual and informational gaps will enrich content and increase reader engagement with this important topic. In this spirit, I encourage you to address all critiques fully.

We thank you, the reviewing editor, and the reviewers for the positive attitudes towards our manuscript. We have filled the gaps highlighted by the reviewers, and believe that our manuscript has been improved by your suggestions.

Response to reviewer #1:

General comments: Timely and needed review highlighting the role of astrocytes in the regeneration of the CNS tissue after trauma. Overall the review is well written well-structured and delivers the message. I support publication.

We thank the reviewer for the positive attitudes and encouraging comments towards our manuscript.

1. Terminology: "activated astrocytes" as used by authors is incorrect. Astrocytes (as well as other glia) are constantly activated in physiological settings by various inputs. In the context of trauma astrocytes become REACTIVE (reactive astrogliosis; reactive astrocytes), see PMID 33589835.

Thank you for this important note. We have updated the manuscript according to the reviewer suggestion and replaced all "activated astrocytes" with "reactive astrocytes".

2. Terminology: "glial scar" does not exist; all scars (in skin or in the brain) are of fibrotic nature; the scar is the replacement of parenchymal cells (which for the CNS are neurones and glia) with stromal cells (pericytes and fibroblasts) which produce matrix. Glial cells erect the perilesional barrier. (see PMID 34164732; doi: 10.1016/B978-0-12-821565-4.00005-5; doi 10.1016/B978-0-12-821565-4.00013-4).

Thank you for pointing this out. Although “glial scar” is common terminology in the field, we agree with the reviewer that it is not technically correct and potentially confusing for the reader, and especially to researchers new to the field. We have therefore updated the manuscript to refer to the perilesional glial barrier according to the reviewer’s suggestion.

3. Terminology: *"The finely branched astrocytic processes contact all cellular components of the CNS and enable a single astrocyte to contact up to 100,000 neurons (16, 17).". Astrocytic processes are classified as branches (primary processes) and leaflets (terminal processes without organelles, which mainly contact synapses - PMID 34479758). In human brain a single protoplasmic astrocyte contacts ~ 2Mil synapses.*

We have revised that statement in page 4 according to the reviewer suggestion.

4. *Probably it might help to the reader if the authors add a paragraph on astrogliopathy; pointing that besides reactivity, in many pathologies astrocytes undergo atrophy and loss of function; even reactive astrocytes can demonstrate either gain or loss of function. Downregulation of glutamate uptake for example is the leading mechanism of secondary neurotoxicity). For details see doi: 10.1016/B978-0-12-821565-4.00005-5; PMID 28805002).*

We have added the suggested paragraph on pages 5 and 6 and revised the manuscript accordingly.

5. *A figure summarizing major mechanisms that can be involved in boosting astrocytes function in order to enhance functional recovery after traumatic brain injuries.*

Thank you for the suggestion; we have added the suggested figure as the new Figure 2.

6. *Tonic GABA inhibition - mid that reactive astrocytes may overexpression MAO-B, hence increasing synthesis and secretion of GABA - look at multiple works of Justin Lee; see also PMID 30877782.*

We have added the suggested paragraph on page 11.

7. *When discussing about Na⁺, may be some references to overall Na homeostasis and signaling in astrocytes can be if value - see PMID 30734296, PMID 26919326.*

We have added the suggested paragraph and references on page 14.

Response to the reviewer #2:

The review is well written and describes fundamental roles of astrocytes. However, some sections of the review are very basic and does not add much to the existing reviews available on the topic. Some sections are not even astrocyte specific. For example, in the section on mechanisms of functional recovery, in the 5-6 pages of the text, there was no specific reference on astrocytes role, which is the topic of this review. Overall, some new information beyond what is described extensively in the past decades and in other reviews are needed to inform the readers about recent advances on astrocyte roles and mechanisms.

We thank the reviewer for their positive attitude towards our manuscript and the valid suggestions. We have revised and trimmed the section of “mechanisms of functional recovery” according to the reviewer’s suggestion.

I have provided some comments to help the authors elevate the content of the review.

- Astrocyte heterogeneity has been an important topic in recent years. A section on this topic needs to be included in the manuscript. There are several recent single cell RNA sequencing and RNA in situ analysis from various research groups (e.g. studies by the Holt and Rowitch labs).

We have now added reference to astrocyte regional heterogeneity at the top of page 5, along with a link to a public scRNA-seq database maintained by the Holt lab.

- In the synaptogenesis section, role of astrocytes in the formation of peri-neuronal net (PNN) should be discussed as it is highly relevant to neuronal activity and plasticity.

We have discussed the suggested topic on page 26.

- For Thrombospondin, it is important to include the seminal work by Barres lab on astroglial production of this critical molecule in the synapse formation section.

Thank you for the suggestion, we have added this on page 25.

- Integrin signaling has been shown to important for the interaction between astrocytes and the ECM in development and after injury, as well as other cell types including endothelial cells. Some discussion on this family of proteins should be included.

We have added some discussion of integrin signaling on page 25.

- Discussion on CSPGs should be expanded as they play important role in CNS repair. There have been major advances in targeting CSPGs more than chondroitinase (e.g. targeting their signaling receptor in axon regeneration, on neural and oligodendrocyte progenitor response after brain/spinal cord injury).

We have added a discussion of CSPG receptors to page 13.

- GFAP in CSF after TBI/concussion is a good biomarker and predictor of functional outcomes. The role of GFAP as a predictive biomarker should be discussed.

We now discuss this topic on page 5.

A minor comment: There is a need for overall proof-reading throughout the manuscript for grammar and typos.

The manuscript has been proofread by a native English speaker and we are confident in the overall quality of the text.

Dear Dr Salman,

Re: JP-SR-2023-284197R1 "Astrocytes in functional recovery following CNS injuries" by Qasim M. Alhadidi
Ghaith A. Bahader
Oiva Arvola
Philip Kitchen
Zahoor A. Shah
Mootaz Salman

I am pleased to tell you that your Symposium Review article has been accepted for publication in The Journal of Physiology, subject to any modifications to the text that may be required by the Journal Office to conform to House rules.

NEW POLICY: In order to improve the transparency of its peer review process, The Journal of Physiology publishes online as supporting information the peer review history of all articles accepted for publication. Readers will have access to decision letters, including all Editors' comments and referee reports, for each version of the manuscript and any author responses to peer review comments. Referees can decide whether or not they wish to be named on the peer review history document.

The last Word version of the paper submitted will be used by the Production Editors to prepare your proof. When this is ready, you will receive an email containing a link to Wiley's Online Proofing System. The proof should be checked and corrected as quickly as possible.

All queries at proof stage should be sent to tjp@wiley.com.

The accepted version of the manuscript is the version that will be published online until the copy edited and typeset version is available. Authors should note that it is too late at this point to offer corrections prior to proofing. Major corrections at proof stage, such as changes to figures, will be referred to the Reviewing Editor for approval before they can be incorporated. Only minor changes, such as to style and consistency, should be made a proof stage. Changes that need to be made after proof stage will usually require a formal correction notice.

Are you on Twitter? Once your paper is online, why not share your achievement with your followers. Please tag The Journal (@jphysiol) in any tweets and we will share your accepted paper with our 22,000+ followers!

Yours sincerely,

Dr Peking Fong
Senior Editor
The Journal of Physiology
<https://jp.msubmit.net>
<http://jp.physoc.org>
The Physiological Society
Hodgkin Huxley House
30 Farringdon Lane
London, EC1R 3AW
UK
<http://www.physoc.org>
<http://journals.physoc.org>

EDITOR COMMENTS:

Reviewing Editor:

Timely and interesting review that should stir some interest.

Senior Editor:

Thank you for addressing comments raised during the previous review of your manuscript.

Both Referees now have had the opportunity to consider the revised version. They both are satisfied with your response to their feedback. In reviewing the incorporated changes, I note the inclusion of deeper exposition suggested by both Referees has markedly enriched the content. This will enhance reader experience, and the extra effort is greatly appreciated.

The Reviewing Editor and I concur on this review's timeliness and its potential to raise interest. I am pleased to recommend acceptance.

REFEREE COMMENTS:

Referee #1:

I am happy with the revised version.

Referee #2:

Authors have adequately addressed my comments and appropriate revisions have been made.

* IMPORTANT NOTICE ABOUT OPEN ACCESS *

To assist authors whose funding agencies mandate public access to published research findings sooner than 12 months after publication, The Journal of Physiology allows authors to pay an open access (OA) fee to have their papers made freely available immediately on publication.

You will receive an email from Wiley with details on how to register or log-in to Wiley Authors Services where you will be able to place an OnlineOpen order.

You can check if your funder or institution has a Wiley Open Access Account here: <https://authorservices.wiley.com/author-resources/Journal-Authors/licensing-and-open-access/open-access/author-compliance-tool.html>.

Your article will be made Open Access upon publication, or as soon as payment is received.

If you wish to put your paper on an OA website such as PMC or UKPMC or your institutional repository within 12 months of publication you must pay the open access fee, which covers the cost of publication.

OnlineOpen articles are deposited in PubMed Central (PMC) and PMC mirror sites. Authors of OnlineOpen articles are permitted to post the final, published PDF of their article on a website, institutional repository, or other free public server, immediately on publication.

Note to NIH-funded authors: The Journal of Physiology is published on PMC 12 months after publication, NIH-funded authors DO NOT NEED to pay to publish and DO NOT NEED to post their accepted papers on PMC.

1st Confidential Review

02-Aug-2023